# Dataset of Partial Analytical Validation of the 1,2-O-Dilauryl-Rac-Glycero-3-Glutaric Acid-(6′-Methylresorufin) Ester (DGGR) Lipase Assay in Equine Plasma

**Laureen Michèle Peters** *[ID] **and Judith Howard**

Clinical Diagnostic Laboratory, Department of Clinical Veterinary Medicine, Vetsuisse Faculty, University of Bern, Länggassstrasse 124, 3012 Bern, Switzerland
* Correspondence: laureen.peters@unibe.ch

**Abstract:** Laboratory assays require analytical validation to prove they are providing accurate results. This dataset describes the partial analytical validation of lipase activity, measured with the 1,2-o-dilauryl-rac-glycero-3-glutaric acid-(6′-methylresorufin) ester (DGGR) lipase assay in equine plasma. Samples with low (approx. 12 U/L), moderately increased (approx. 79 U/L), and markedly increased lipase activity (approx. 298 U/L) were chosen. Linearity was assessed in samples of ascending dilution prepared by mixing samples with low and high lipase activity in different proportions. Repeatability or intra-assay replication was evaluated by measuring each level in 25 replicates within the same run. Reproducibility or inter-assay replication was calculated by measuring each level in five replicates on five consecutive days. The assay was linear in the range of 12–298 U/L ($R^2 = 0.9998$) with a <2.3% deviation from the calculated value at any point. Within-run coefficients of variation were 4.43%, 0.69%, and 1.00% for the low, medium, and high samples, respectively. Between-run coefficients of variation were 3.57%, 1.42%, and 1.16%, respectively. To our knowledge, these are the first published data on the analytical validation of the DGGR lipase assay in horses, which may be of interest to veterinary clinical pathologists and equine clinicians measuring DGGR lipase in equine blood for diagnostic and research purposes.

**Dataset:** Partial analytical validation data of the DGGR lipase assay in equine plasma. DOI: 10.6084/m9.figshare.19549339. Repository name: Figshare. Direct URL to data: https://doi.org/10.6084/m9.figshare.19549339.v1.

**Keywords:** horse; pancreas; enzyme; linearity; repeatability; reproducibility

## 1. Summary

The measurement of lipase activity using the 1,2-o-dilauryl-rac-glycero-3-glutaric acid-(6′-methylresorufin) ester (DGGR) substrate is routinely performed for the diagnosis of pancreatitis in canine and feline blood samples in many veterinary diagnostic laboratories; however, little is published to date about the use of this assay in horses. One study demonstrated high tissue specificity for DGGR lipase in the pancreatic tissue of horses and found that hyperlipasemia was common in horses with gastrointestinal disease [1]. We recently showed that DGGR lipase was increased above reference intervals in 30% of horses with colic and that hyperlipasemia above 2x the upper reference interval was associated with large bowel displacement or torsion, surgical treatment, strangulating bowel disease, and non-survival [2]. As the DGGR assay has not been previously validated for use in equine plasma samples, we performed partial analytical validation as part of the said study. Briefly, DGGR lipase activity was measured in the leftover plasma from blood samples taken for biochemical testing as part of the routine diagnostic workup of

client-owned horses presented to a veterinary teaching hospital. Samples with normal (within reference intervals), moderately, and markedly increased DGGR lipase activity were selected. Measurements were performed using a commercially available assay (LIPC, Roche) on a routine biochemistry analyzer (Cobas c501, Roche). Linearity, repeatability, and reproducibility were evaluated following the American Society for Veterinary Clinical Pathology's Quality Assurance and Laboratory Standards guidelines [3]. A summary of these results has been published in the original article [2]; here, the methods and the resulting dataset are presented in detail.

The objectives of these investigations were to provide analytical validation data for the use of the DGGR lipase assay in equine plasma, in order to justify its validity for application in equine practice and research.

To our knowledge, these are the first published data on the analytical validation of the DGGR lipase assay in horses. Our data will be of interest to veterinary clinical pathologists and equine clinicians measuring DGGR lipase in equine blood for diagnostic purposes. Furthermore, researchers investigating the diagnostic and prognostic value of DGGR lipase activity for different diseases in horses can refer to our validation data.

## 2. Data Description

The data on the partial analytical validation of the DGGR lipase assay on equine plasma samples in [2] are deposited in a public repository, available at https://doi.org/10.6084/m9 .figshare.19549339.v1 (last accessed 17 January 2023), in Microsoft Excel table format with three different sheets, one for each of the experiments described below, labelled accordingly.

### 2.1. Linearity Study

The first column "BARCODE" and the second column "sample name" are the laboratory's unique internal sample identifiers and sample name, respectively. The third column "Pool" describes the dilution pools from which the sample originated. The next three columns describe the serum indices, as measured with the analyzer. Column G "DGGR lipase" contains the numerical measurements of DGGR lipase activity, expressed in U/L. In the following three columns H-J, the calculated means of all four repeated measurements of the same pool are listed, followed by the theoretically expected value (calculated from the proportions of low and high samples in the pool, as described in Methods), as well as the % deviation of the measured mean from the expected value. Lastly, columns K, L, and M contain the results of the regression analysis. The assay was linear in the range of 12–298 U/L ($R^2$ = 0.9998) with excellent analyte recovery (<2.3% deviation from the calculated value at any point). The regression equation was Measured value = $-0.3832 + 0.9945 \times$ Expected value. For the visual inspection of linearity, the measured mean values are plotted against the expected values in Figure 1.

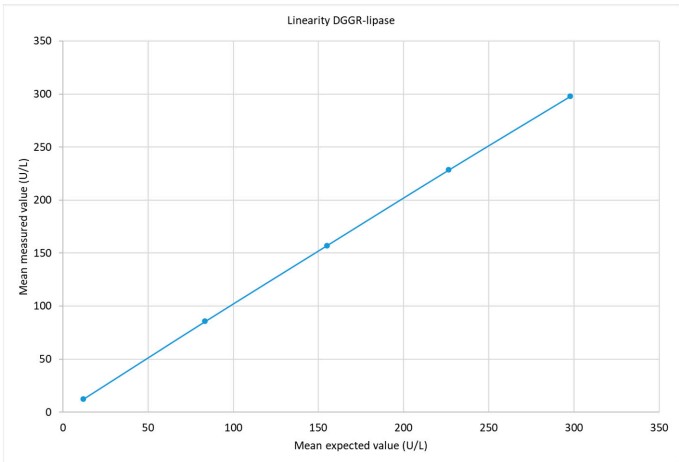

**Figure 1.** Linearity of the 1,2-o-dilauryl-rac-glycero-3-glutaric acid-(6′-methylresorufin) ester (DGGR) lipase assay demonstrated by plotting mean measured results (y-axis) against mean expected values (x-axis).

## 2.2. Repeatability Study (Intra-Assay Replication)

The first column "BARCODE" and the second column "sample name" are the laboratory's unique internal sample identifiers and sample name, respectively. The third column "Pool" describes from which of the three plasma pools, namely low, mid, or high pool, the sample originated. The next three columns describe the serum indices, as measured by the analyzer. Column G "DGGR lipase" contains the numerical measurements of DGGR lipase activity, expressed in U/L. In the following six columns H-M, the median, interquartile range, range, mean, standard deviation, and coefficient of variation of all 25 repeated measurements of the same pool are listed. For an overview, these are summarized in Table 1.

**Table 1.** Summary of results of the repeatability study.

| Pool | N | DGGR Lipase | | | |
| | | Median (IQR) (U/L) | Range (U/L) | Mean $\pm$ SD (U/L) | CV (%) |
| --- | --- | --- | --- | --- | --- |
| Low | 25 | 12 (12–12) | 11–13 | 11.88 $\pm$ 0.5260 | 4.43 |
| Mid | 25 | 78 (78–78) | 77–79 | 78.04 $\pm$ 0.5385 | 0.69 |
| High | 25 | 297 (295–300) | 293–303 | 297.44 $\pm$ 2.9732 | 1.00 |

CV, coefficient of variation; IQR, interquartile range; SD, standard deviation.

## 2.3. Reproducibility Study (Inter-Assay Replication)

The first column "BARCODE" is the laboratory's unique internal sample identifier. The second column "sample name" is the sample name, whereby D1 designates measurements performed on day 1, D2 represents repeat measurements on day 2, and so forth. The third column "Pool" describes from which of the three plasma pools of DGGR lipase activity, namely low, mid, or high, the sample originated. The next three columns describe the serum indices, as measured by the analyzer. Column G "DGGR lipase" contains the numerical measurements of DGGR lipase activity, expressed in U/L. Column H contains the mean of all five repeat measurements of the same level on the same day. In the following six columns (I-N), the median, interquartile range, range, between-day mean, standard deviation, and coefficient of variation of all 25 repeated measurements of the same pool on all 5 different days are listed. For an overview, these are summarized in Table 2.

**Table 2.** Summary of results of the reproducibility study.

| Pool | N | DGGR Lipase | | | |
| | | Median (IQR) (U/L) | Range (U/L) | Mean $\pm$ SD (U/L) | CV (%) |
| --- | --- | --- | --- | --- | --- |
| Low | 5 | 12.8 (12.60–13.05) | 12.0–13.2 | 12.76 $\pm$ 0.4561 | 3.57 |
| Mid | 5 | 80.2 (79.85–81.35) | 78.8–81.8 | 80.44 $\pm$ 1.1437 | 1.42 |
| High | 5 | 299.2 (298.7–302.85) | 297.8–306.6 | 300.84 $\pm$ 3.5026 | 1.16 |

## 3. Methods

Blood samples were collected from client-owned horses that were presented to the Swiss Institute of Equine Medicine of the Vetsuisse Faculty, University of Bern, Switzerland, as part of the routine diagnostic workup for spontaneously occurring diseases. Leftover heparinized plasma was stored at $-25$ °C for up to three weeks, which is within the manufacturer's reported stability of DGGR lipase. Three samples corresponding to low (approx. 12 U/L; within published reference intervals of <21 U/L [1]), moderately increased (approx. 79 U/L, or roughly 4x upper reference level (URL)), and markedly increased DGGR lipase activity (approx. 298 U/L; >10x URL) were selected, originating from a 10-year-old mare of unknown breed, a 19-year-old warmblood gelding, and a 15-month-old Swiss warmblood colt.

Lipase activity was measured on heparinized equine plasma using a commercially available DGGR assay (LIPC, Cat. 03029590322, Roche Diagnostics, Switzerland) on an

automated biochemistry analyzer (Cobas c501, Roche diagnostics, Switzerland) according to the manufacturer's instructions at the Clinical Diagnostic Laboratory, Department of Clinical Veterinary Medicine, Vetsuisse Faculty, University of Bern, Switzerland. The assay was calibrated using the manufacturer's standard calibrator (Calibrator for Automated Systems, Cat. 10759350360, Roche diagnostics, Switzerland). Two levels of controls (PreciControl ClinChem Multi 1 and Multi 2, Cat. 05117003160 and 05117216160, Roche diagnostics, Switzerland) were run daily prior to sample measurements with CVs of 2.31% and 1.67% for the low control (43 U/L) and high control (95 U/L), respectively. External quality assurance was run on a monthly basis and was considered "excellent" for the DGGR lipase assay during the period of measurement. According to the manufacturer's datasheet, this assay is not influenced by interferences by icterus, hemolysis, or lipemia up to indices of 60, 1000, or 2000, respectively. No data or instrument alarms were observed during the analyses.

Analytical validation was performed following the ASVCP guidelines for quality assurance in veterinary clinical pathology [3].

Linearity was evaluated using plasma samples of low and high DGGR lipase activities, with dilution samples prepared by mixing the low and high samples in different proportions as follows, measured in 4 replicates each:

- Level 1: Low sample;
- Level 2: 3 parts low sample, 1 part high sample;
- Level 3: 2 parts low sample, 2 parts high sample;
- Level 4: 1 part low sample, 3 parts high sample;
- Level 5: High sample.

Repeatability and reproducibility were assessed by evaluating intra- and inter-assay coefficients of variation (CVs). For intra-assay precision, the samples of all three activity levels were measured in 25 replications during the same run. For inter-assay variability, the samples of all three levels were measured five days in a row, with five replications on each day. To avoid multiple freeze–thaw cycles, samples were prepared on day 1 and stored at 4 °C until analysis (up to 4 days), which is within the manufacturer's reported stability of DGGR lipase.

Statistical analysis was performed using commercial software (MedCalc Statistical Software version 20.104, MedCal Software Ltd., Ostend, Belgium) and MS Excel. Linearity was visually assessed by plotting expected values and mean measured values; linearity and line of best fit were calculated with regression analysis. For repeatability, the means, standard deviations (SDs), and CVs were calculated; for reproducibility, the means of the replicate measurements of the different activity levels within a single day were used to calculate SDs and CVs between runs.

**Author Contributions:** Conceptualization, L.M.P. and J.H.; methodology, L.M.P. and J.H.; formal analysis, J.H.; investigation, L.M.P.; resources, J.H.; data curation, J.H.; writing—original draft preparation, L.M.P.; writing—review and editing, J.H.; visualization, J.H.; supervision, J.H.; project administration, L.M.P.; funding acquisition, J.H. All authors have read and agreed to the published version of the manuscript.

**Funding:** This research received no external funding.

**Institutional Review Board Statement:** Ethical review and approval were waived for this study, as all blood samples were leftover samples originally collected for routine diagnostic workup.

**Informed Consent Statement:** Consent was obtained from horse owners for all diagnostic procedures and for the use of leftover biological material for research purposes.

**Data Availability Statement:** Datasets generated and analyzed during this study can be found at https://doi.org/10.6084/m9.figshare.19549339.v1 (accessed on 17 January 2023).

**Acknowledgments:** The authors would like to thank Nicole Zufferey for her help in preparing and measuring the samples.

**Conflicts of Interest:** The authors declare no conflict of interest.

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
