# Peer review of "Dataset of Partial Analytical Validation of the 1,2-O-Dilauryl-Rac-Glycero-3-Glutaric Acid-(6′-Methylresorufin) Ester (DGGR) Lipase Assay in Equine Plasma"

_data, 2022_

Round 1

Reviewer 1 Report

The Authors describe the validation test of the instrumentation in use in the structure in which they work. the topic is interesting and the methodology correct.

The paper in my opinion can be published in present form: it is a description of the metodology for the validation of the lipase activity in horses. The technique is described in a straight-forward manner, for linearity, reproducibility and repeatability.

Author Response

We would like to thanks this reviewer for taking the time to review our manuscript and for their favorable comments.

Reviewer 2 Report

In this article, the authors present the first existing data on analytical validation of the 1,2- 10 o-dilauryl-rac-glycero-3-glutaric acid-(6’-methylresorufin) ester (DGGR) lipase assay in horses, which may be of interest to veterinary clinical pathologists and equine clinicians. The manuscript complies with all the requirements of the target journal. Only one minor issue needs to be addressed (please check the attachment).

Author Response

We would like to thanks this reviewer for taking the time to review our manuscript and for their favorable comments.

Regarding the addition of sources of error and noise in the manuscript: We have added that the storage conditions of our samples are within the manufacturer's reported stability, and had already described how we avoided freeze-thaw cycles to minimize errors due to sample stability. We have also added that this assay takes part in our monthly external QA program and has performed excellently during the study period, to highlight that major systematic bias is unlikely. Furthermore, we have added that no data or instrument alarms were raised during the runs pertinent to this assay. This assay is not influenced by interferences through icterus, hemolysis, or lipemia up to very high levels, which we have now also added to the text, and from our data table it is visible that no indices in our samples exceed (or even come close to) these limits. It is also stated in the data sheet that the assay is not influenced by common drugs, and only Waldenström's macroglobulinemia is listed as potential endogenous error source, which was not the case in any of our study horses, so we decided not to specifically mention this. We hope that with this additional information, we could demonstrate that we have limited potential errors as best we could, and hope that this adequately addresses this reviewer's concerns. As there is no discussion section in this type of paper, we saw no other place to address and expand on these potential issues within the manuscript.

Reviewer 3 Report

This study is well planned and researched about 1,2-o-dilauryl-rac-2 glycero-3-glutaric acid-(6’-methylresorufin) ester (DGGR) lipase 3 assay in equine plasma. 

The researhers stated that this is the first published data on analytical validation of the DGGR-lipase assay in horses, which may be of interest to veterinary clinical pathologists and equine clinicians measuring DGGR-lipase in equine blood for diagnostic and research purposes.

When evaluated in this respect, I believe that the data obtained will make important contributions to the studies to be carried out in this field.

Author Response

(The authors gave the same response as above.)
